# Unraveling surface and bulk dynamics of iron(III) molybdate during oxidative dehydrogenation using operando and transient spectroscopies

Leon Schumacher[1], Mariusz Radtke[1], Jan Welzenbach[1] & Christian Hess [1 ✉]

Iron(III) molybdate ($Fe_2(MoO_4)_3$) is a commercial catalyst for the oxidative dehydrogenation (ODH) of methanol, but it has recently been shown to be relevant for other substrates as well. Despite its commercial use, a detailed mechanistic understanding of $Fe_2(MoO_4)_3$ catalysts at the surface and in the bulk has been lacking, largely hampered by the lack of suitable spectroscopic methods, directly applicable under reaction conditions. Using propane ODH as an example, we highlight the potential of operando Raman and impedance spectroscopy combined with transient IR spectroscopy, to identify surface active sites and monitor the hydrogen transfer and oxygen dynamics. By comparison with the behavior of reference compounds ($MoO_3$, $MoO_x/Fe_2O_3$) a mechanistic model is proposed. The presence of iron greatly influences the reactivity behavior via oxygen diffusion but is moderated in its oxidative capacity by surface $MoO_x$. Our approach directly elucidates fundamental properties of $Fe_2(MoO_4)_3$ of general importance to selective oxidation catalysis.

[1] Technical University of Darmstadt, Department of Chemistry, Eduard-Zintl-Institut für Anorganische und Physikalische Chemie, Peter-Grünberg-Str. 8, 64287 Darmstadt, Germany. ✉email: christian.hess@tu-darmstadt.de

$Fe_2(MoO_4)_3$ is a well-known catalyst used commercially for the oxidative dehydrogenation (ODH) of methanol to formaldehyde[1,2] but it has recently been shown to have also potential for other oxidation reactions, such as ethanol ODH[3–5]. In these reactions, $Fe_2(MoO_4)_3$ shows even higher activities than the commonly used supported vanadia ($VO_x$) catalysts[6] and is therefore very promising. However, the mode of operation of $Fe_2(MoO_4)_3$ catalysts has been the subject of numerous kinetic and structural studies[7,8] as summarized in excellent reviews[5,9,10]. More recently, there has been renewed interest in $Fe_2(MoO_4)_3$ catalysts, and new insights into the bulk dynamics were provided by operando X-ray diffraction (XRD) and X-ray absorption spectroscopy (XAS)[11,12]. Despite the progress made there is still a distinct lack of mechanistic understanding of the (sub)surface processes in $Fe_2(MoO_4)_3$ catalysts under reaction conditions and their relation to bulk properties, including the role of iron[5,7,9,13–16]. A problem commonly encountered with such catalysts is that the conversion, even at low temperatures, is very high, leading to additional effects (such as mass transport) that make the mechanistic analysis more complex. One way to approach this is to consider alkane oxidation reactions, such as propane ODH, which in addition to being of great interest for technical applications, might also facilitate the evaluation of fundamental and transferable properties of $Fe_2(MoO_4)_3$, because the reaction has lower conversions than alcohol ODH at similar temperatures[6,17]. For a detailed mechanistic understanding, it will be of particular importance to further define the role of molybdenum and iron, which is facilitated by the use of reference materials, including $MoO_3$ (characterized by low conversions and high selectivities) and $Fe_2O_3$ (shown to be a total oxidation catalyst with high conversions and $CO_x$ yields)[14,18,19].

In this contribution, we present a combined operando and transient spectroscopic study on the reaction mechanism of propane ODH over $Fe_2(MoO_4)_3$ catalysts. Our focus is on combining methodological approaches of general applicability to selective oxidation reactions, such as modulation-excitation (ME-) IR spectroscopy[20], operando impedance spectroscopy on powder samples[21], and dedicated resonance enhancement using multi-wavelength Raman spectroscopy[22], aiming at a profound understanding of the (sub)surface and the bulk processes, as well as their relevance for catalysis. Our findings serve as a fundamental basis for the mechanistic investigation of oxidation reactions with much higher conversions.

## Results

**Catalytic activity**. The $Fe_2(MoO_4)_3$ was synthesized using co-precipitation and extensive structural characterization was performed, as described elsewhere[3]. A brief summary of important characterization data for this study is given in the Supporting Information SI (see Supplementary Table 1). Additional X-ray photoelectron spectroscopy (XPS) data for our $Fe_2(MoO_4)_3$ batch is provided in the SI (see Supplementary Fig. 1 and Supplementary Table 2), showing a higher surface concentration of molybdenum (Mo/Fe ratio of 2.52) than would be expected from the stoichiometry, which is in good agreement with previous studies[4,5,13,16]. Analysis of the Fe $2p_{3/2}$ photoemission (see Supplementary Fig. 2) reveals that most of the iron is present as $Fe^{3+}$ (711 eV). Besides, a small shoulder due to $Fe^{2+}$ (709 eV) is detected[23], which may be indicative of some $FeMoO_4$ being present as a side phase in addition to the already detected $Fe_2O_3$ by Mössbauer spectroscopy[3].

The reactivity behavior of $Fe_2(MoO_4)_3$ in propane ODH was determined within the temperature range 25–550 °C (see Fig. 1a). The corresponding consumption of oxygen is given in the SI (see Supplementary Fig. 3) and shows that oxygen is never fully consumed.

The conversion of propane increases exponentially with increasing temperature and constantly stays above that of the empty reactor, while the selectivity is initially around 40%, showing a plateau, and then starts to decline continuously at around 320 °C, following the increase in conversion. To understand the reaction mechanism, two temperatures were chosen for operando measurements, i.e., 320 and 500 °C. At 320 °C the conversion is 0.91% with a selectivity of 35.5% while at 500 °C the conversion significantly increases to 6.94% but the selectivity decreases to 18.8%. The conversions at both temperatures are significantly below those during the ODH of methanol or ethanol[3,24,25], which eliminates additional kinetic effects such as mass transport, enabling a detailed mechanistic investigation of the reaction mechanism and fundamental surface/bulk properties of the material transferable to other reactions. While the chosen temperatures represent different aspects of the reactivity behavior, the exponential decay of the selectivity–conversion plot (see Fig. 1b) indicates that the reaction mechanism stays similar within the temperature range covered in this study. The selectivity–conversion plot for a sequence reaction (like the selective/total oxidation of propane is proposed to be) shows an exponential behavior, while its exact shape depends on the ratio of the rate constants. A switch in reaction mechanisms usually leads to a disruption of this exponential course, which cannot be observed here[26]. Furthermore, the temperature-dependent molar product distributions (see Supplementary Fig. 4) show that all products produced at 500 °C ($C_3H_6$, CO, and $CO_2$) increase exponentially with the temperature, as expected for similar pathways leading to their formation. The amount of $CO_2$ increases the most, indicating a significant increase in oxygen mobility necessary for this amount of total oxidation of propane. This is in contrast to methanol oxidation[3,9,27,28], for example, where not all side products show an exponential increase with increasing temperature, but rather go through maxima, indicating a change of the underlying reaction mechanism, which is not detected for our propane data. The observed behavior strongly suggests a similar ODH reaction mechanism at 320 and 500 °C, allowing operando experiments to be performed at low conversion with higher selectivity and at high conversion.

**Transient spectroscopy**. Starting with the structural dynamics at the surface, transient IR spectroscopy was applied during reaction conditions by using modulation-excitation (ME-) DRIFTS coupled with phase-sensitive detection (PSD), which allows active species to be discriminated from spectator species, as described in detail in the SI. A general description of the MES/PSD procedure and how the results described here were obtained based on time-resolved spectra is also given in the SI (see Supplementary Figs. 5, 6, and Supplementary Discussion 1)[20,29,30]. Fig. 2 shows the PSD spectra in a constant propane and pulsed oxygen flow at 320 °C with a 30° phase resolution. The assignment to the propane gas phase over KBr after PSD treatment are given in the SI (see Supplementary Fig. 6). There are no ME-IR spectra for 500 °C due to increased thermal emission, which leads to very noisy spectra.

The PSD spectra of $Fe_2(MoO_4)_3$ show mostly gas-phase contributions from propane and one additional peak between 990 and 1050 cm$^{-1}$ with a maximum at 1020 cm$^{-1}$. This peak was previously assigned to the terminal Mo=O stretching vibration of amorphous molybdenum oxide ($MoO_x$) supported on $SiO_2$ and in reduced $MoO_3$[31–33], indicating the presence of a molybdenum-rich surface layer in $Fe_2(MoO_4)_3$. Note that this feature has not been observed previously in Raman spectra of bulk oxides, and its participation in the reaction is accessible here only due to the increased sensitivity of the modulation-excitation spectroscopy

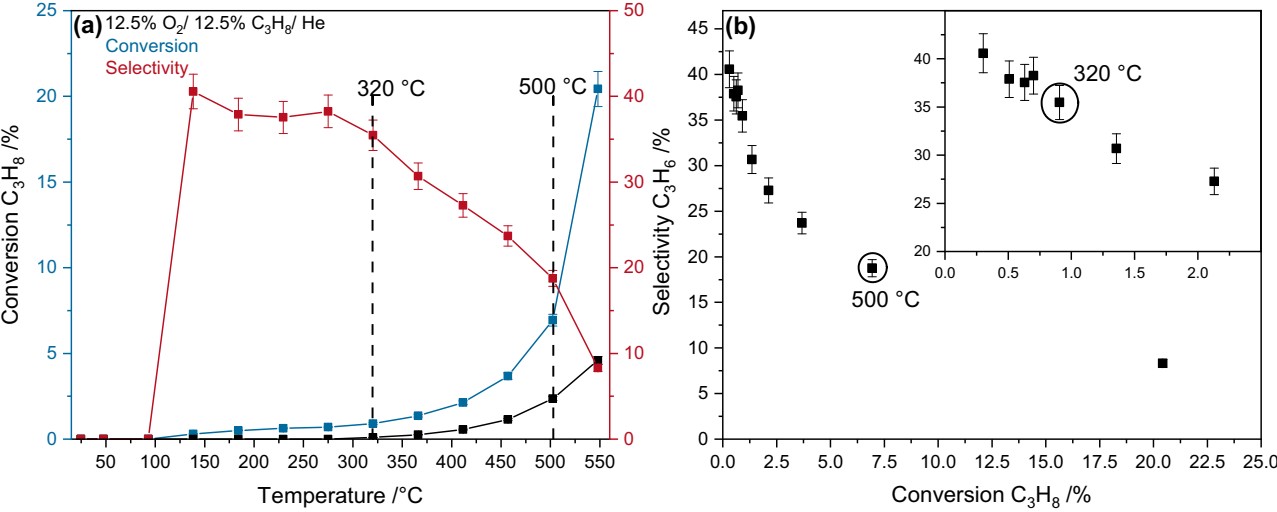

**Fig. 1 Catalytic performance of Fe₂(MoO₄)₃ during propane ODH. a** Temperature-dependent reactivity behavior of $Fe_2(MoO_4)_3$ in a feed of 12.5% $O_2$/ 12.5% $C_3H_8$/He (total flow: 40 ml/min). The temperatures chosen for the operando experiments are indicated and the background conversion caused by the reactor is indicated by the black line. **b** Propylene selectivity as a function of conversion. The inset shows an enlarged view of the low conversion region for clarity. The temperatures at which spectroscopic experiments were performed are indicated. The error bars were produced as a percentage of the areas obtained from the GC by error propagation.

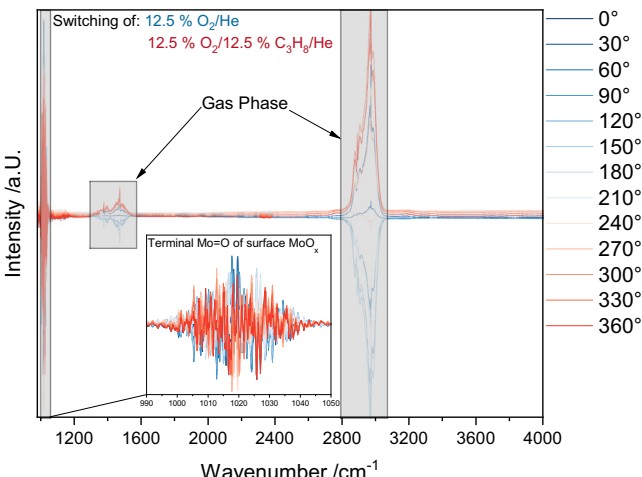

**Fig. 2 Transient IR spectroscopic behavior of Fe₂(MoO₄)₃.** The spectra are based on PSD spectra with a 30° phase resolution during constant propane and pulsed oxygen flow at 320 °C. The gas-phase contributions from propane are highlighted and the inset shows an enlarged view of the terminal Mo = O stretching region. For details see text.

(MES)/PSD approach as a change in the Mo = O signal cannot be observed in the time domain (see Supplementary Fig. 5)[29,30]. The presence of amorphous $MoO_x$ on the surface of $Fe_2(MoO_4)_3$ was previously proposed based on ex-situ scanning transmission electron microscopy (STEM) and electron energy loss spectroscopy (EELS) analysis[13] and is consistent with our XPS results (see Supplementary Table 2), but was not measured directly under reaction conditions and evidenced an actively participating species in ODH reactions. The small concentration of surface $MoO_x$ (likely less than a couple of monolyers[5,34,35]) in combination with the low conversion explains the small intensity of the Mo = O signal and underlines the need for a highly sensitive approach as provided by ME-DRIFTS coupled with PSD. A transient V = O signal was observed previously in supported vanadia catalysts as a result of a fast hydrogen transfer from propane to the catalyst surface and subsequent

regeneration[20]. As there is no indication of a change in mechanism with temperature (see Fig. 1), a similar behavior may also be operative in the case of the (supported) molybdenum surface layer of iron molybdate catalysts.

In addition to the ME-DRIFTS results, operando UV-Vis spectra (see Supplementary Fig. 7) reveal small contributions from d–d transitions between 600 and 800 nm caused by the presence of reduced $Mo^{4/5+}$ species under reactive conditions, which are in good agreement with the reduction of Mo = O groups during the hydrogen transfer from propane to $MoO_x$, further indicating the reduction and participation of Mo=O groups during the reaction[32]. The amount of d–d transitions also increases concurrently with the increase in conversion between 320 and 500 °C while no additional structural dynamics is detected, further underlining the occurrence of the same reaction mechanism.

**Operando spectroscopy.** The two wavelengths at which Raman spectroscopy was performed are highlighted in the UV-Vis spectra (see Supplementary Fig. 7), as the choice of the excitation wavelength can greatly influence the Raman intensity and the depth of information. At 514 nm, the absorption is small and is expected to be mostly caused by oligomerized $MoO_x$ on the catalyst surface[32], which may undergo a selective resonance enhancement. However, this is not observed in the Vis-Raman (514 nm excitation) spectra (see Supplementary Fig. 8 and Supplementary Table 3 for assignments), which can be explained by the high depth of penetration of the visible laser, thus gathering information from the bulk and thereby covering the small contribution from surface $MoO_x$ species. Hence, no structural dynamics can be detected at this wavelength for $Fe_2(MoO_4)_3$, indicating that the bulk structure is not significantly changed during the reaction at either temperature. In contrast, in the case of the UV-Raman spectra (at 385 nm excitation) the absorption is much higher (see Supplementary Fig. 7), giving rise to resonance enhancements mainly caused by transitions in the molybdate[36]. Fig. 3 depicts operando Raman spectra of $Fe_2(MoO_4)_3$ at 320 and 500 °C recorded in a feed of 12.5% $O_2$/12.5% $C_3H_8$/He compared to oxidative conditions (12.5% $O_2$/He). Based on the increased absorption, the UV wavelength has a smaller depth of penetration

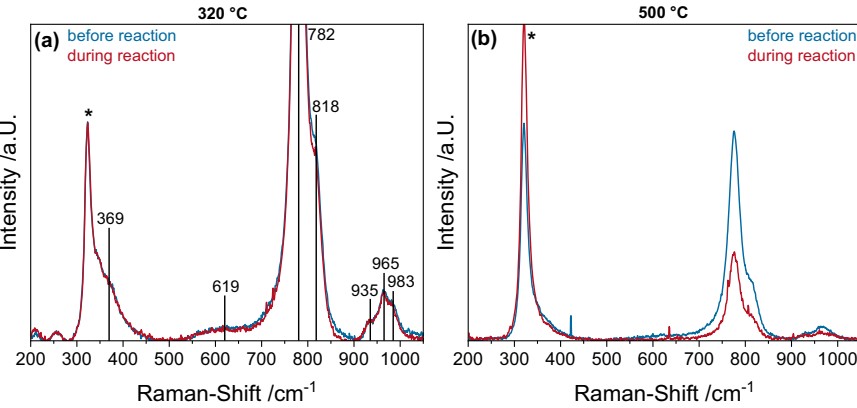

**Fig. 3 Operando UV-Raman spectra (at 385 nm excitation) of Fe₂(MoO₄)₃.** The spectra were recorded in a feed of 12.5% O₂/12.5% C₃H₈/He compared to oxidative conditions (12.5% O₂/He) at **a** 320 °C and **b** 500 °C. The asterisk (*) marks a peak resulting from the CaF₂ window used.

compared to the Vis, leading to a difference in the depth over which information is gathered by UV-Raman compared to Vis-Raman spectroscopy, which is more focused on the subsurface of the material.

The spectra measured at 320 °C show only very slight but reproducible changes in the intensity of the antisymmetric $MoO_4$ stretching modes at 782 and 818 cm$^{-1}$ (see Fig. 3a), indicating that the $MoO_4$ in the $Fe_2(MoO_4)_3$ lattice is reduced during the reaction. At 500 °C these changes are much more pronounced (see Fig. 3b), consistent with the increase in conversion, indicating a significant transport of oxygen from the $Fe_2(MoO_4)_3$ subsurface lattice to the surface of the catalyst, facilitating the ODH reaction. This is also in good agreement with the proposition that the mechanism is the same over the investigated temperature range, as higher temperatures facilitate the formation of oxygen vacancies in the $Fe_2(MoO_4)_3$ lattice and subsequent oxygen diffusion to the surface. As a consequence, there is more oxygen available for the reaction, thus increasing the reaction rate including the over-oxidation of propane to $CO_x$, which is in good agreement with the observed selectivities. The results from operando/transient ME-DRIFT, UV-Vis spectroscopy, and Raman spectroscopy indicate that surface $MoO_x$ facilitates the initial hydrogen transfer from propane to the catalyst, which is the rate-determining step, while lattice oxygen of $Fe_2(MoO_4)_3$ acts as the oxidizing agent. So far, the discussion has focused on the role of surface $MoO_x$ and the involvement of lattice oxygen. However, the influence of iron and oxygen transport properties is likely to be important, all the more so as the catalyst is characterized by a very small surface area and large particles (as shown previously for this sample)[3], emphasizing the importance of the bulk properties.

To enhance the understanding of the transport properties within $Fe_2(MoO_4)_3$, we applied an operando powder impedance spectroscopic approach, developed in our group[21]. For comparison, $MoO_3$ and $Fe_2O_3$ loaded with $MoO_x$ (1 Mo/nm²) were measured as reference samples, to better determine the influence of the Fe and Mo concentration on the catalytic activity. For best possible comparability, the $Fe_2O_3$ sample had a similar particle size and surface area to the $Fe_2(MoO_4)_3$ one[3]. Operando impedance spectra of the three samples at 320 °C and of $MoO_3$ at 500 °C are shown in the SI (see Supplementary Figs. 9 and 10).

Since the conversions are low at 320 °C for all three samples, the degree of reduction is also small, and due to the low temperatures, the conductivity of the material is not sufficient for good-quality impedance spectra. This leads to a high degree of noise, which makes the interpretation of the spectra difficult (see Supplementary Fig. 9). Therefore, we focused on the 500 °C

impedance spectra, which allow access to mechanistic information due to the much higher oxygen mobility and hence conductivity of the materials. Based on all the previous evidence in the kinetic and spectroscopic results, it seems likely that the reaction mechanism stays the same between both temperatures and the effects detected only increase concurrently with the conversion, the assumption that we can focus on the 500 °C impedance spectra seems justified. Based on the impedance data of $Fe_2(MoO_4)_3$ at 500 °C under oxidative and reactive conditions (see Fig. 4a), a detailed equivalent-circuit model analysis was performed, enabling a physical interpretation of each electric component (see Supplementary Figs. 11 and 12, and Supplementary Discussion 2). The region left of the dashed line is not shown as it is composed of electrode resistance and inductance due to the material skin effect (see discussion in the SI). When the gas phase is switched from oxidative to reactive conditions, the overall conductivity of the material increases, which is in good agreement with its reaction-induced reduction (see Fig. 3 and Supplementary Fig. 7) and the increased mass transport of oxygen ions. As explained in detail in the SI, the impedance data suggests oxygen ion mass transport from additional phases ($Fe_2O_3$, $MoO_x$, $FeMoO_4$) to be too small to be detected by XRD, as well as hydrogen and water transport in the molybdenum-rich and more defective/porous surface regions of $Fe_2(MoO_4)_3$[5,21,37]. This is in good agreement with the operando UV-Raman and UV-Vis data, providing a clear indication of the relevance of oxygen transport through the material. Furthermore, the resistance element R5 (see Supplementary Fig. 11) indicates hydrogen transport through the $Fe_2(MoO_4)_3$ surface layer in agreement with ME-DRIFTS data. In addition, taking into account water reduction (in the sub-circuit containing R3, R4, C4, and W), significantly increases the quality of the equivalent-circuit fit compared to a fit based on oxygen transport phenomena only (see Supplementary Fig. 12). In comparison, the impedance spectra of $MoO_x/Fe_2O_3$ shown in Fig. 4b depict none of the additional features displayed by $Fe_2(MoO_4)_3$ but behave close to an ideal R-C equivalent circuit, indicating the importance of oxygen ion transport only, without any additional phases. A contribution of $MoO_x$ is not detected for the $MoO_x/Fe_2O_3$ sample, as the surface loading of $MoO_x$ is expected to be much smaller (1 Mo/nm²) compared to the near stoichiometric presence of Mo close to the $Fe_2(MoO_4)_3$ surface. The $MoO_3$ impedance spectrum (see Supplementary Fig. 10) shows a poor signal-to-noise ratio with very significant resistances, indicating low conductance.

Comparing the conductance of the three samples to their conversion reveals a correlation. Importantly, the resistance of all samples decreases when the gas phase is switched from oxidative

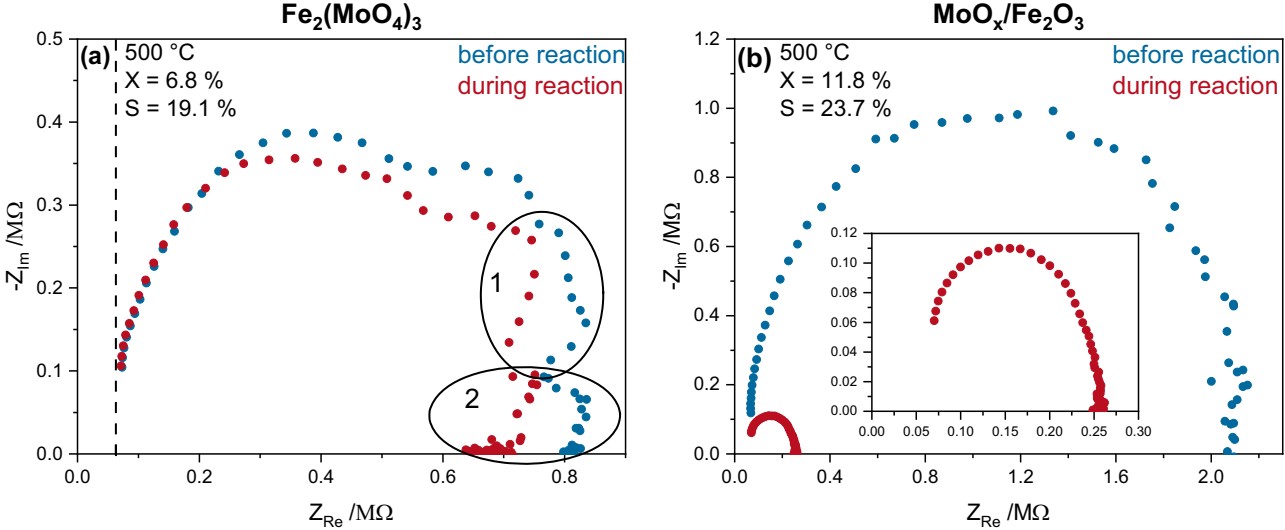

**Fig. 4 Operando impedance spectroscopic results.** The spectra show **a** Fe$_2$(MoO$_4$)$_3$ and **b** MoO$_x$/Fe$_2$O$_3$ at 500 °C in a feed of 12.5% O$_2$/12.5% C$_3$H$_8$/He compared to oxidative conditions (12.5% O$_2$/He). The dashed line marks the region of bulk electrode resistance and the numbered circles highlight the regions in which the Fe$_2$(MoO$_4$)$_3$ impedance spectrum differs from that of the MoO$_x$/Fe$_2$O$_3$. The operando impedance spectroscopic results for MoO$_3$, Fe$_2$(MoO$_4$)$_3$, and 1.0MoO$_x$ + Fe$_2$O$_3$ at 320 °C and MoO$_3$ at 500 °C are shown in the SI (see Supplementary Figs. 9 and 10).

to reactive conditions, indicating an increased oxygen mobility due to the reducing gas atmosphere[38]. The lowest conversion of 0.4% is observed for crystalline MoO$_3$, which is accompanied by a resistance in the megohm region, but cannot be quantified further due to the high noise. Fe$_2$(MoO$_4$)$_3$ and MoO$_x$/Fe$_2$O$_3$ show lower resistances of ~700 and ~250 kΩ and higher conversions of 6.8% and 11.8%, respectively, exhibiting a clear correlation between iron content, oxygen mobility, and conversion. This suggests that a higher concentration of iron is the driving factor for higher conversions, while the presence of molybdenum seems to be important for C–H bond breakage, which is supported by the similar selectivities of Fe$_2$(MoO$_4$)$_3$ and MoO$_x$/Fe$_2$O$_3$, despite their differences in conversion.

In summary, applying an operando impedance spectroscopic approach developed in our group to the Fe$_2$(MoO$_4$)$_3$ system allowed direct investigation of the influence of the catalyst composition on the oxygen transport properties and therefore conversion.

## Discussion
Based on the operando and transient spectroscopic results presented above, a reaction mechanism is proposed (see Fig. 5). First, a hydrogen atom is abstracted from propane via a Mo = O group (shown by ME-DRIFTS, see Fig. 2 and Supplementary Figs. 5 and 6) of the molybdenum-rich surface and transferred quickly to the catalyst surface. The reduced molybdenum can then be seen in the d–d transitions observable in the operando UV-Vis spectra (see Supplementary Fig. 7). The hydrogen atom then diffuses through the surface layer of the catalyst, where hydroxyl and/or water can be formed, as shown by the equivalence circuit of operando impedance spectroscopy (see Supplementary Fig. 11 and Supplementary Discussion 2). Similarly, the second hydrogen atom is abstracted and bound to lattice oxygen, eventually leading to the formation of water, which desorbs from the catalyst into the gas phase (see Supplementary Figs. 11, 12 and Supplementary Discussion 2), thereby leaving an oxygen vacancy, which can then be replenished by subsurface oxygen via oxygen transport to the surface (seen by the conductivity and oxygen transport increase in the impedance spectra) and gas-phase oxygen. As a result of the reaction, there is an overall reduction of the material, both at the molybdenum-enriched surface (ME-DRIFTS, UV-Vis) and in

the subsurface (UV-Raman, impedance; see Figs. 2, 3, 4 and Supplementary Figs. 8, 9 and 11), which facilitates the formation of FeMoO$_4$, which is more conductive and can be seen clearly in the operando impedance spectra (see Supplementary Fig. 11). Additionally, an increased amount of iron in the lattice increases the oxygen mobility and therefore the conversion but due to the high oxygen mobility, Fe$_2$O$_3$-based catalysts often lead to total oxidation[5], which was also shown by the significant increase in conductivity in the impedance spectra (Fig. 4b) due to the quick transport of oxygen. In contrary, MoO$_3$ has been demonstrated to be an excellent selective oxidation catalyst but shows low conversion in ODH reactions[15], which could be explained by its low oxygen mobility (see Fig. 4). Fe$_2$(MoO$_4$)$_3$ combines these two properties effectively by increasing the oxygen mobility compared to MoO$_3$, and in addition, by forming a thin layer of amorphous MoO$_x$ on the surface of the catalyst, greatly increasing the selectivity for ODH in comparison to Fe$_2$O$_3$. The green and red triangles indicate an increase or decrease of the respective value mentioned, this is, it describes the influence of an increase of the iron or molybdenum content on the conversion and selectivity.

In summary, we applied multiple operando and transient spectroscopies to investigate the mechanism of propane ODH over Fe$_2$(MoO$_4$)$_3$. Our findings suggest the M=O groups of MoO$_x$ to be an active site that is responsible for the abstraction of hydrogen from C–H bonds, which is commonly described as the rate-determining step in propane ODH[6]. Multiple effects within the subsurface/bulk contribute to a further enhancement of the catalytic activity, including percolation of hydrogen in the subsurface, the formation of water, subsequent phase transformation, and regeneration by oxygen diffusion and gas-phase oxygen. Hereby, the concentration of iron is crucial for the oxygen mobility within the material and can influence the catalytic activity substantially. Owing to the structure of Fe$_2$(MoO$_4$)$_3$, both high conversion, due to copious amounts of iron, and good selectivity, due to the interaction of surface MoO$_x$ with the gas phase, can be achieved. By applying multiple operando and transient spectroscopic approaches under working conditions, we were able to directly access fundamental properties of Fe$_2$(MoO$_4$)$_3$ catalysts that were previously a matter of debate or only observed indirectly. The method of our approach is readily transferable to other oxide catalysts and reactions, while our

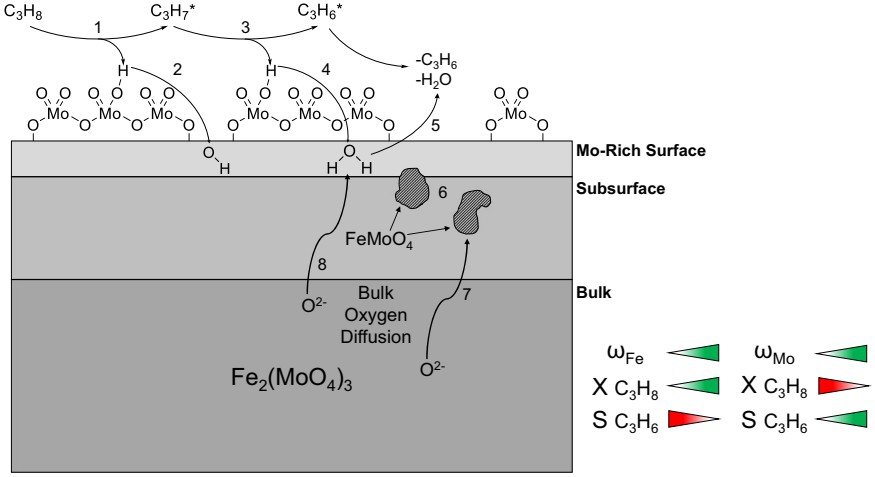

**Fig. 5 Mechanistic scheme of the ODH of propane over Fe$_2$(MoO$_4$)$_3$ catalysts.** H atom abstraction from propane (1, 3) is followed by hydrogen transfer to lattice oxygen (2, 4), leading to water formation (5) and leaving an oxygen vacancy. As a result of the reaction, the (sub)surface is reduced, facilitating the formation of FeMoO$_4$ (6), which can be replenished by oxygen diffusion from the bulk (7, 8). The influence of the content (ω) of iron and molybdenum on the catalytic performance (conversion: X; selectivity: S) is illustrated in the bottom right.

findings are expected to serve as a basis for a detailed mechanistic understanding of the mode of operation of selective oxidation catalysts.

## Methods

**Catalyst preparation**. The synthesis of Fe$_2$(MoO$_4$)$_3$ by co-precipitation[3]. Briefly, iron nitrate nonahydrate (Merck, ≥98%) and ammonium heptamolybdate (Merck, ≥99%) were dissolved separately in demineralized water. The aqueous iron solution was then added dropwise to the molybdate solution while it was stirred. The mixed solutions were stirred for three more hours to complete the precipitation process. The precipitate was first filtered and washed with demineralized water and ethanol and then dried overnight at 100 °C in air. Finally, the powder was calcined in air at 500 °C for 10 h.

For the reference samples, iron (III) oxide (Sigma Aldrich, particle size <5 μm, >96%) was loaded by incipient wetness impregnation with ammonium heptamolybdate tetrahydrate. For that, the iron(III) oxide was dispersed in an aqueous solution of ammonium heptamolybdate tetrahydrate (Fluka, puriss. p. a. >99%) under continuous stirring. The dispersion was dried at 90 °C overnight, followed by calcination at 600 °C for 12 h.

MoO$_3$ was synthesized by calcination of recrystallized ammonium heptamolybdate tetrahydrate (Fluka, puriss. p. a. >99%) at 600 °C for 12 h.

**Catalytic testing**. Catalytic testing was performed in a CCR1000 reaction cell using 120 mg of catalyst. The sample was first dehydrated for 1 h at 365 °C in 12.5% O$_2$/He (40 ml$_n$/min). The catalyst was then cooled to 25 °C, exposed to 12.5% O$_2$/12.5% C$_3$H$_8$/He with a total flow rate of 40 ml$_n$/min, and then heated to 550 °C in 45 °C steps, staying at each temperature for 1 h. The gas-phase composition was analyzed continuously using gas chromatography (GC, Agilent Technologies 7890B). The GC is equipped with a PoraPlotQ and a Molsieve column as well as a thermal conductivity detector (TCD) and a flame ionization detector (FID) in series. The setup is connected through a twelve-way valve. One chromatogram is measured every 29 min, resulting in two chromatograms for each temperature, which were averaged. The pressure before and after the GC was monitored to correct the detected areas for pressure fluctuations.

**Table 1 Relative sensitivity factors (RSFs) for quantification of elemental concentrations from XP spectra.**

| Element and level | Fe 2p$_{3/2}$ | Mo 3d | O 1s | C 1s |
|---|---|---|---|---|
| RSF | 7.96 | 9.79 | 2.50 | 1.00 |

**X-ray photoelectron spectroscopy**. X-ray photoelectron spectroscopy (XPS) was carried out on an SSX 100 ESCA spectrometer (Surface Science Laboratories Inc.) employing a monochromatic Al Kα X-ray source (1486.6 eV) operated at 9 kV and 10 mA; the spot size was approximately 1 mm × 0.25 mm. The base pressure of the analysis chamber was <10$^{-8}$ Torr. Survey spectra (eight measurements) were recorded between 0 and 1100 eV with 0.5 eV resolution, whereas detailed spectra (30 measurements) were recorded with 0.05 eV resolution. To account for sample charging, the C 1s peak of ubiquitous carbon at 284.4 eV was used to correct the binding-energy shifts in the spectra. Atomic concentrations were calculated using the relative sensitivity factors (RSFs) given in Table 1.

**Diffuse reflectance UV-Vis spectroscopy**. Diffuse reflectance (DR) UV-Vis spectra were recorded on a Jasco V-770 UV-Vis spectrometer. Dehydrated BaSO$_4$ was used as the white standard. For each experiment, 120 mg of catalyst was put in the commercially available reaction cell (Praying Mantis High Temperature Reaction Chamber, Harrick Scientific) equipped with transparent quartz glass windows. The Harrick cell was calibrated separately to ensure the same temperatures in both reaction cells used. Operando spectra were measured at 320 and 500 °C under oxidizing (12.5% O$_2$/He) and reactive (12.5% C$_3$H$_8$/12.5% O$_2$/He) conditions, after 1 h of dehydration in 12.5% O$_2$/He at 365 °C (total flow rate: 40 ml$_n$/min). Before measuring each spectrum, the samples were pretreated in the respective gas phase for 30 min, to ensure a steady state.

**Visible Raman spectroscopy**. Visible (Vis) Raman spectroscopy was performed at 514 nm excitation, emitted from a Cobolt Fandango diode laser (Hübner Photonics). The light was focused onto the sample, gathered by an optical fiber, and dispersed by a transmission spectrometer (Kaiser Optical, HL5R). The dispersed

Raman radiation was subsequently detected by an electronically cooled charge-coupled device (CCD) detector (–40 °C, 1024 × 256 pixels). The spectral resolution was 5 cm$^{-1}$ with a wavelength stability of better than 0.5 cm$^{-1}$. For Raman experiments, 120 mg of catalyst was filled into a CCR1000 reactor (Linkam Scientific Instruments) equipped with a quartz window (Linkam Scientific Instruments). The laser power at the sample location was 4 mW. Data analysis of the Raman spectra included cosmic ray removal and an auto-new dark correction. Operando spectra were measured at 320 and 500 °C under oxidizing (12.5% $O_2$/He) and reactive (12.5% $C_3H_8$/12.5% $O_2$/He) conditions, after 1 h of dehydration in 12.5% $O_2$/He at 365 °C (total flow rate: 40 ml$_n$/min). Before measuring each spectrum, the samples were pretreated in the respective gas phase for 30 min, to ensure a steady state.

**UV-Raman spectroscopy**. UV-Raman spectroscopy was performed at an excitation wavelength of 385 nm generated by a laser system based on a titanium sapphire (Ti: Sa) solid-state laser pumped by a frequency-doubled Nd: YAG laser (Coherent, Indigo). The fundamental wavelength is frequency doubled to 385 nm using a $LiB_3O_5$ crystal. The light is focused onto the sample, and the scattered light is collected by a confocal mirror setup and focused into a triple-stage spectrometer (Princeton Instruments, TriVista 555)[39]. Finally, the Raman contribution is detected by a CCD (2048 × 512 pixels) cooled to –120 °C. The spectral resolution of the spectrometer is 1 cm$^{-1}$. For Raman experiments, 120 mg of catalyst was placed in a CCR1000 reactor (Linkam Scientific Instruments) equipped with a $CaF_2$ window (Korth Kristalle GmbH). The laser power at the location of the sample was 5 mW. Data processing included cosmic ray removal and background subtraction. Operando spectra were measured at 320 and 500 °C under oxidizing (12.5% $O_2$/He) and reactive (12.5% $C_3H_8$/12.5% $O_2$/He) conditions, after 1 h of dehydration in 12.5% $O_2$/He at 365 °C (total flow rate: 40 ml$_n$/min). Before measuring each spectrum, the samples were pretreated in the respective gas phase for 30 min, to ensure a steady state.

**Modulation excitation (ME)-DRIFTS**. ME-DRIFT spectroscopy was performed using a modified apparatus that was already for steady-state DRIFTS experiments[40,41]. For each experiment, 120 mg of catalyst was used.

We used the rapid scan mode of Bruker's spectrometer software OPUS 7.2. Spectra were measured from 850 to 3800 cm$^{-1}$ with a resolution of 0.5 cm$^{-1}$, an aperture of 8 mm, and a mirror speed of 40 kHz. A Valco Instruments 4/2 valve (Model E2CA, version ED), communicating with the Vertex 70, is used to rapidly switch between different gas feeds, which are controlled *via* digital mass flow controllers (Bronkhorst).

As gases we used $C_3H_8$ (Westfalen, 3.5), $O_2$ (Westfalen, 5.0), and helium (Westfalen, 5.0). One measurement series consisted of 20 periods (20 gas-phase switches), each of which had a duration of 360 s and consisted of 240 spectra. For one spectrum, five consecutive interferograms were averaged, so that a new spectrum was acquired every 1.5 s.

As background the catalyst spectrum itself was used, after 1 h of dehydration at 365 °C in 12.5% $O_2$/helium atmosphere and a 10 min treatment at 320 °C in one of the reaction gases for conventional ME-DRIFTS (12.5% $O_2$ or 12.5% $C_3H_8$ in helium). The flow was kept constant at 100 ml$_n$/min during the pretreatment and experiment.

During ME-DRIFTS, a flow of either 12.5% $C_3H_8$ or 12.5% $O_2$ in helium was kept constant over the sample, while the other feed gas was pulsed over the sample. The temperature during all modulation-excitation experiments was kept at 320 °C. To remove the gas-phase contribution, we subtracted gas-phase spectra over KBr from each recorded DRIFT spectrum. To exclude the possibility of intensity fluctuations over multiple periods, we checked the intensity profile at three distinct wavenumbers, representative of the background, an adsorbate peak, and a gas-phase peak, but detected no absolute intensity changes over multiple periods that could influence the Fourier transformation.

To obtain phase-sensitive spectra, the time-resolved 3D spectral data was converted from the time to the phase domain. For an overview, the resolution of phase spectra was chosen to be 30°, whereas mechanistic insights were obtained using a resolution of 1°. The main operation of phase-sensitive detection (PSD) is a Fourier transformation according to[42]

$$I_{\widetilde{\nu}}(\varphi) = \frac{2}{T} \int_0^T I_{\widetilde{\nu}}(t) \cdot \sin\left(2\pi f t + \varphi\right) dt$$

where $I(t)$ is the time-dependent intensity at one specified wavenumber ($\widetilde{\nu}$) that is convoluted with the sine function, representing the modulation of the external parameter (e.g., the gas-phase concentration), thus forming $I(\varphi)$, the phase-resolved intensity. The frequency of the external modulation is $f$, and 0 and $T$ represent the times at which the considered dataset begins and ends, respectively. To obtain a complete phase-resolved spectrum, this procedure is repeated for every wavenumber. By varying $\varphi$ from 0 to 360° with a chosen resolution and repeating the steps above, the complete phase-resolved dataset is created.

The strong noise in the signal of the ME-DRIFT spectra (see Fig. 2) is caused by the low $MoO_x$ concentration and low conversion as the intensity after the Fourier transform depends on the intensity change when switching the gas-phases.

**Impedance spectroscopy**. Potentiostatic electrochemical impedance spectra (p-EIS) were acquired in a two-electrode system using a BioLogic VSP potentiostat/galvanostat operated in the 1 MHz to 1 Hz range with 60 mV amplitude and 20 measurement points per decade acquired in triplicate with a potential of 0.05 V versus a reference of +0.771 V ($Fe^{3+}$ to $Fe^{2+}$). The positive reduction potential was referenced against the standard hydrogen electrode (SHE). Impedance spectra were recorded after placing 120 mg of the catalyst in a commercial CCR1000 cell (Linkam Scientific Instruments, UK), equipped with a PTFE (polytetrafluoroethylene) plate with two holes for the copper electrodes, as described previously[21]. In this context, we also performed experiments with gold electrodes (Alfa Aesar, UK, 99.999%), where no influence of the electrode material on the electrochemical output was observed with the exception of a parasitic potential iR drop, arising from the peculiarities of the cell assembly. Its compensation was performed manually in the EC-Lab v. 11.33 (BioLogic, France) acquisition software prior to the actual measurement. Before each measurement, to allow for equilibration, the sample was kept at 320/500 °C for about 30 min under oxidative (12.5% $O_2$/He) or reactive (12.5% $O_2$/12.5% $C_3H_8$/He) conditions, respectively (total flow rate: 40 ml$_n$/min). This procedure ensures that the measurements take place in a stationary state as verified by considering the temporal evolution, which did not show any significant changes during the measurement. Raw spectra were validated by applying the Kramers–Kronig relations, which deviate from an ideal behavior by ca. 8%, with 11% being the benchmark for discarding the measurement, meaning that the real and imaginary parts of the experimental spectra overlap with say, the imaginary spectral points calculated from the real part by applying Hilbert transforms. Before measuring each spectrum, the samples were

pretreated in the respective gas phase for 30 min, to ensure a steady state.

## Data availability

The datasets generated during and/or analyzed during the current study are not publicly available but are available from the corresponding author on reasonable request.

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

## Acknowledgements

The authors acknowledge Karl Kopp for XPS measurements and spectral analysis, and Katrin Hofmann for the synthesis of $Fe_2(MoO_4)_3$. This work was supported by the Deutsche Forschungsgemeinschaft (DFG, HE 4515/11-1). Leon Schumacher, Jan Welzenbach, and Christian Hess gratefully acknowledge financial support by the CRC 1487.

## Author contributions

L.S. performed the majority of the experimental work, designed the measurement protocols, analyzed most of the data, and was a major contributor in writing this paper. M.R. performed the detailed equivalent-circuit analysis of the operando impedance spectra. J.W. supplied the samples. C.H. wrote the grants, provides the experimental equipment, supervised the project, and was a major contributor in discussing the results and writing this paper.

## Funding

## Competing interests

The authors declare no competing interests.
