## [Peer Review File · Communications Chemistry]

Reviewers' comments:

Reviewer #1 (Remarks to the Author):

This article reports on new data to unravel the surface and bulk species, and their connection, as outlined in Figure 5 of the article, during the oxidative dehydrogenation of light alkanes over $\text{Fe}_2(\text{MoO}_4)_3$ making use of operando and transient spectroscopy. The authors report on the role of the surface species, and the surface-subsurface-bulk dynamics in the overall performance of the catalyst materials. The methods employed are X-ray photoelectron spectroscopy, Raman spectroscopy, measured with two different lasers (i.e., 385 and 514 nm), UV-Vis DRS, as well as modulation excitation infrared spectroscopy (ME-IR). This approach is powerful and allows to obtain new insights.

Unfortunately, there are some caveats in the article which makes that I recommend major revisions. First of all, I believe that because the bulk is mentioned as crucial next to XPS, Raman, ME-IR and UV-Vis DRS also XRD and/or XAS should be employed; preferentially also under operando conditions. There is experimental evidence missing to fully support the scheme made in Figure 5. Secondly, I am of the opinion that the data are not well discussed and written down; that becomes evident when reading the part of the ME-IR; which I believe should be discussed in terms of the time and phase domains and the two sets of spectra should be presented and based this a translation has to be made towards what is surface and gas-phase contributions; and it is currently far from trivial to understand the M=O region as shown in the insert in the figure. Related to this it would be good also to expand the number of results figures and better link the different results obtained as measured at the two different temperatures. There is some rework needed to connect the dots as provided by the different analytical methods presented. I do not yet arrive to the same conclusions derived by the authors in this study.

Summarizing, original and new data are presented under relevant reaction conditions, which are certainly worth publishing, but the overall scientific message is for me not yet clear, nor is the presentation of the data made in such a way that the reader can grasp all the information presented, and culminating in Figure 5.

Reviewer #2 (Remarks to the Author):

Schumacher et al. report the use of $\text{Fe}_2(\text{MoO}_4)_3$ catalyst for oxidative dehydrogenation (ODH) of propane, including extensive operando characterization using DRIFTS, Raman, UV-vis and impedance spectroscopy.

The paper is well written, generally easy to follow, and the content is very interesting and the results and procedures at a very high level, which provides significant new insights to the $\text{Fe}_2(\text{MoO}_4)_3$ catalyst. In terms of catalytic performance, the $\text{Fe}_2(\text{MoO}_4)_3$ catalyst is rather poor for propane ODH compared to e.g. supported vanadia catalysts, but this is not the main purpose of the study. There are some minor details that needs to be clarified before the paper can be recommended for publication:

Specific comments:

TOC graphic: What do the green or red triangles represent? The same in Fig. 5? I am not able to figure out what they mean.

P.4: Fe₂O₃ “shown to be burner”? Maybe “shown to be a total oxidation catalyst”?

P. 5: Give a quick summary of the bulk characterization methods from the ref. 3: BET surface area, bulk Mo/Fe ratio from ICP, XRD phases detected and quantified. This would help the reader of this manuscript.

P5: Fe₂O₃ is Fe³⁺, so should not contribute to Fe²⁺ detected by XPS.

P5: Was the reactor tested for homogeneous or steel surface reactions at 500 to 550 C? And for the related SI Fig. S2 what was the oxygen mol flow. For complete oxidation of propane the reaction feed is still under-stoichiometric in oxygen, so it is good to confirm that not all oxygen was consumed. Furthermore, since it is a flow reactor the amount should be mole flow, or?

P9: For Fig. 3, Fig. S3 and Fig. S4 how long time was the catalyst exposed to reaction conditions before acquiring the “during reaction” spectra (red curves)? And was the catalyst in a steady state or were changes occurring with time?

P.10: What was the BET surface area of the MoO_x/Fe₂O₃ and Fe₂(MoO₄)₃ samples?

P. 15: Missing n in iron.

P. 16: Was the temperature of the sample in the Praying Mantis cell measured. It is well known that the sample temperature is significantly lower than the cell temperature.

Reviewer #3 (Remarks to the Author):

Review report on manuscript:

Unraveling Surface and Bulk Dynamics of Fe₂(MoO₄)₃ during Oxidative Dehydrogenation using Operando and Transient Spectroscopies

by Leon Schumacher, Mariusz Radtke, Jan Welzenbach, Christian Hess

In this paper the authors use a number of different operando techniques to study a Fe₂(MoO₄)₃ catalyst while exposed to propane/oxygen mixtures including temperature-dependent reactivity measurements, transient IR spectroscopy, UV-Raman spectroscopy, and impedance spectroscopy. Based on the combined experimental evidence the authors suggest a model with MoO_x sites at the Mo-rich surface,

which are responsible for the abstraction of H-atoms from propane, while subsurface FeMoO₄ increase the oxygen mobility to ensure rapid regeneration of the reduced MoO_x sites.

Altogether, I enjoyed reading the paper, which is well-written and there is – as far as I can judge – solid experimental evidence for the conclusions given. I therefore recommend publication of the paper and only have limited critics which is given below:

Minor critics:

1) In the abstract the authors write: “...we highlight the potential of operando Raman and impedance spectroscopy combined with transient IR spectroscopy...” and later in the introduction it is written: “..., might also facilitate the evaluation of novel methodical approaches...”. Finally in the conclusion it is written: “The method of our approach is readily transferable to other oxide catalysts and reactions, while our findings are expected to serve as a basis for a detailed mechanistic understanding of the mode of operation of selective oxidation catalysts.”

I do not understand this claim. Transient IR spectroscopy is not new. The authors (and other authors) used and published work with this method before. For example, ref. 20 is recent paper using this method. The same is true for the other operando methods used. To use 3 well-established operando techniques is hardly a “novel methodical approach”. I suggest that the authors put less focus on this and adjust the language.

2) In the paper the authors imply that the “additional x-ray photoelectron spectroscopy data” are shown in table S1 of the SI. When reading this I expected to see raw data (i.e. spectra), but only the results are given. This give readers no chance of checking the raw data themselves. I think it is good practice always to give both raw data and conclusion and suggest that the overview spectrum is included in the SI.

3) On p. 6 the authors write: “...the exponential decay of the selectivity–conversion plot (see Figure 1b) indicates that the reaction mechanism stays similar within the temperature range covered in this study.” I am not familiar with selectivity-conversion plots and how to interpret them, and maybe not all readers will be. Therefore, I encourage the readers to explain this better.

4) The sentence on p. 6 “..., which allows active species to be discriminated from spectator species, as described in detail in the SI. 20”, is misleading. This is not explained in the SI. Instead, it is explained in the SI. I encourage the authors to discuss this in the SI also.

5) On p. 7 the authors write: “Note that this feature has not been observed previously in vibrational spectra of bulk oxides, and is accessible here only due to the increased (surface-) sensitivity of the modulation excitation spectroscopy (MES)/PSD approach.” This is a claim without any proof. To demonstrate this they need to show spectra with and without modulation.

6) On the top of p. 8 the authors write: “reactions. The small concentration of surface MoO_x...”. How small is this concentration? What is the experimental evidence for the small concentration? Please specify.

Referee #1

We thank the referee for her/his comments.

First of all, I believe that because the bulk is mentioned as crucial next to XPS, Raman, ME-IR and UV-Vis DRS also XRD and/or XAS should be employed; preferentially also under operando conditions. There is experimental evidence missing to fully support the scheme made in Figure 5.

While we agree that the methods mentioned by the referee are not sufficient to support the scheme in Figure 5, we point out that operando impedance spectroscopy, which plays an important part in this study (Figures 4 and S5-8) but was not mentioned by the referee, is even better suited than XRD or XAS to directly detect the mass transport through the catalyst under reaction conditions. This approach was established recently for the bulk investigation of In_2O_3 catalysts bulk during reverse water gas shift reaction (10.1002/anie.202209388) and was shown to be highly sensitive to the gas environment. A detailed description of impedance spectroscopy and its use for investigating semiconducting materials is given in the literature, as cited in our study (10.1021/acsaem.9b01965, 10.1149/1945-7111/ab77a0, 10.1039/d1ra03785d). Besides, this approach can provide indirect hints towards the presence of the FeMoO_4 phase due to its better oxygen mobility (and therefore conductivity) (10.1016/0254-0584(84)90099-3), which is not possible via XRD, as previous studies on this material proposed this phase to be nanocrystalline (10.1039/d0cp01506g).

Secondly, I am of the opinion that the data are not well discussed and written down; that becomes evident when reading the part of the ME-IR; which I believe should be discussed in terms of the time and phase domains and the two sets of spectra should be presented and based this a translation has to be made towards what is surface and gas-phase

contributions; and it is currently far from trivial to understand the M=O region as shown in the insert in the figure.

For details on the ME-DRIFTS data acquisition and analysis, we provide a thorough experimental description as well as the reference to two of our publications. For the assignments of gas-phase peaks, commonly used positions of gaseous propane are used. To emphasize this point, we added two Figures to the revised SI: first, the requested Figure of spectra in the time and phase domain, respectively, to show their transformation, and second, a Figure of the phase-resolved spectra of KBr, highlighting the gas-phase contributions.

Regarding the Mo=O region, no precise assignments are given but only the assignment to an Mo=O vibration, which is similarly to the V=O region located at higher wavenumbers than for crystalline MoO₃ (V₂O₅), indicative of amorphous Mo=O surface species. The assignment of this region to Mo=O vibrations of amorphous MoO_x is supported by literature results (10.1039/B107046K, 10.1039/b107012f, 10.1023/B:CATL.0000029521.80714.8d)

Related to this it would be good also to expand the number of results figures and better link the different results obtained as measured at the two different temperatures. There is some rework needed to connect the dots as provided by the different analytical methods presented. I do not yet arrive to the same conclusions derived by the authors in this study.

As described above, we added additional result Figures to the SI. To improve the clarity of the mechanistic findings, we expanded the mechanistic discussion in the revised version of the manuscript, more strongly emphasizing the connections between the results from the different methods by highlighting the relevant Figures, which were used as a basis for the respective mechanistic conclusion.

The observed reactivity behavior strongly suggests similar reaction mechanisms at 320 and 500 °C. As a confirmation, we used spectroscopic results at both temperatures (UV-Vis, Vis-Raman, and UV-Raman spectroscopy), while ME-DRIFT spectra were accessible at 320 °C only, due to the temperature limitations of DRIFTS, and impedance spectra were accessible at 500 °C only, due to the transport limitations at 320 °C, leading to significant noise.

Referee #2

We thank the referee for her/his comments.

TOC graphic: What does the green or red triangles represent? The same in Fig. 5? I am not able to figure out what they mean.

We thank the referee for her/his comments and agree that the description is incomplete. The symbols represent an increase or decrease of the respective value, that is, an increased amount of iron increases the conversion and decreases the selectivity, while an increased amount of molybdenum decreases the conversion and increases the selectivity. We added an explanation to the text of the revised manuscript.

P.4: Fe₂O₃ “shown to be burner”? Maybe “shown to be a total oxidation catalyst”?

We agree and changed the text accordingly.

P. 5: Give a quick summary of the bulk characterization methods from the ref. 3: BET surface area, bulk Mo/Fe ratio from ICP, XRD phases detected and quantified. This would help the reader of this manuscript.

While we provided a reference to our previous study [3], which includes an extensive characterization of the catalyst, for clarity, we added a Table to the revised SI (Table S1), summarizing the most important characterization results.

P5: Fe₂O₃ is Fe³⁺, so should not contribute to Fe²⁺ detected by XPS.

This misunderstanding is likely due to an ambivalent phrasing. What we mean is that the small Fe²⁺ contribution is likely caused by FeMoO₄, which is an additional phase present

besides Fe_2O_3 , which was detected by Mössbauer spectroscopy. For clarity, we rephrased the section in the revised manuscript.

P5: Was the reactor tested for homogeneous or steel surface reactions at 500 to 550 °C? And for the related SI Fig. S2 what was the oxygen mol flow. For complete oxidation of propane the reaction feed is still under-stoichiometric in oxygen, so it is good to confirm that not all oxygen was consumed. Furthermore, since it is a flow reactor the amount should be mole flow, or?

We tested the reactor over a wide temperature range for homogeneous and steel surface reactions and added the results to the catalysis section. Furthermore, the oxygen traces determined by GC were added to the SI, showing that oxygen is not fully consumed due to the low propane conversions. The reactor itself is a flow reactor, but the GC tube gathering the gas sample is not and is characterized by a fixed volume and gathering time. In combination with the pressure measurements we perform, an amount instead of a mole flow can be calculated.

P9: For Fig. 3, Fig. S3 and Fig. S4 how long time was the catalyst exposed to reaction conditions before acquiring the “during reaction” spectra (red curves)? And was the catalyst in a steady state or was changes occurring with time?

The samples were first dehydrated for 1h in 12.5% O_2/He at 366 °C, before setting the reaction temperature (Note: a new sample was used for the 320 and 500 °C measurements, respectively) and waiting for 30 minutes before recording the spectra in oxidizing conditions. Afterwards, reaction conditions were set and after 30 minutes of equilibration, the spectrum was recorded, ensuring a steady state. For clarity, we expanded the description in the experimental section of the revised manuscript.

P.10: What was the BET surface area of the $\text{MoO}_x/\text{Fe}_2\text{O}_3$ and $\text{Fe}_2(\text{MoO}_4)_3$ samples?

We added the BET surface area values of the samples to the revised SI (Table S1).

P. 15: Missing n in iron.

We thank the referee for his comment and fixed the typo accordingly.

P. 16: Was the temperature of the sample in the Praying Mantis cell measured. It is well known that the sample temperature is significantly lower than the cell temperature.

We calibrated the temperatures in both cells separately. The set temperature value was varied accordingly for each cell, so that the sample temperature in both cells was the same.

Referee #3

We thank the referee for her/his comments.

1) In the abstract the authors write: "...we highlight the potential of operando Raman and impedance spectroscopy combined with transient IR spectroscopy..." and later in the introduction it is written: "..., might also facilitate the evaluation of novel methodical approaches...". Finally in the conclusion it is written: "The method of our approach is readily transferable to other oxide catalysts and reactions, while our findings are expected to serve as a basis for a detailed mechanistic understanding of the mode of operation of selective oxidation catalysts."

I do not understand this claim. Transient IR spectroscopy is not new. The authors (and other authors) used and published work with this method before. For example, ref. 20 is recent paper using this method. The same is true for the other operando methods used. To use 3 well-established operando techniques is hardly a "novel methodical approach". I suggest that the authors put less focus on this and adjust the language.

While we agree with the referee, that none of the methods in isolation is new, operando impedance spectroscopy on powder samples has been published only very rarely, and especially the combination of all three methods was, to the best of our knowledge, never applied to a catalyst enabling the detailed differentiation between the surface (ME-DRIFTS), the subsurface (UV-Raman, with resonance enhancement), and the direct observation of transport through the bulk (impedance spectroscopy). Is it only the method combination, which enables us to conclude on a mechanistic picture with great level of detail. In any case, we adjusted some statements in the introduction of the revised manuscript to decrease the focus on the novelty.

2) In the paper the authors imply that the "additional x-ray photoelectron spectroscopy data" are shown in table S1 of the SI. When reading this I expected to see raw data (i.e. spectra), but only the results are given. This give readers no chance of checking the raw data themselves. I think it is good practice always to give both raw data and conclusion and suggest that the overview spectrum is included in the SI.

Because the XPS data is given and discussed in detail in the referenced characterization study of this material we showed only the Fe 2p emission because of its relevance for the discussion. For clarity, we also added the overview spectrum of Fe₂(MoO₄)₃ to the revised SI.

3) *On p. 6 the authors write: "...the exponential decay of the selectivity–conversion plot (see Figure 1b) indicates that the reaction mechanism stays similar within the temperature range covered in this study." I am not familiar with selectivity-conversion plots and how to interpret them, and maybe not all readers will be. Therefore, I encourage the readers to explain this better.*

We agree with the referee and added a short description and a reference explaining these plots to the revised version of the manuscript.

4) *The sentence on p. 6 "..., which allows active species to be discriminated from spectator species, as described in detail in the SI. 20", is misleading. This is not explained in the SI. Instead, it is explained in the SI. I encourage the authors to discuss this in the SI also.*

It is true that the origin of the discrimination between active and spectator species was described in detail in the SI of reference 20 and not in the SI of this manuscript. While detailed description of why the MES approach discriminates between active and observer species can be found in the literature (10.1039/C9RE00011A, 10.1016/j.ces.2007.06.009), we added a short description of this behavior together with additional data on the modulation-excitation experiments to the revised SI.

5) *On p. 7 the authors write: "Note that this feature has not been observed previously in vibrational spectra of bulk oxides, and is accessible here only due to the increased (surface-) sensitivity of the modulation excitation spectroscopy (MES)/PSD approach." This is a claim without any proof. To demonstrate this they need to show spectra with and without modulation.*

Our claim is justified by the time-resolved spectra (shown as Figure S5 in the revised SI) as well as by the previous use of ME-DRIFTS. The surface sensitivity of ME-DRIFTS is well established in the literature and we added references to support that point. We emphasize that with our statement we refer to ME-DRIFTS, as the MES/PSD approach itself, as correctly pointed out by the referee, does not enhance the surface sensitivity *per se*, but the overall sensitivity towards weak signals if they participate in the reaction. Therefore, to identify the actively participating surface MoO_x sites in the Fe₂(MoO₄)₃ catalyst, we consider ME-DRIFTS as the (currently) best choice. To improve the clarity, we modified the corresponding sentence on page 8 of the revised manuscript.

6) *On the top of p. 8 the authors write: "reactions. The small concentration of surface MoO_x...". How small is this concentration? What is the experimental evidence for the small concentration? Please specify.*

Since the co-precipitated Fe₂(MoO₄)₃ is a commonly used catalyst, previous analysis is available in that regard. Previous EDX and XPS measurements have showed increased surface concentrations of molybdenum and an amorphous MoO_x surface layer was proposed (10.1016/j.susc.2015.11.010, 10.1021/jp5081753, 10.1002/cctc.202101219). The presence of the Mo=O signal at 1020 cm⁻¹ in ME-DRIFTS confirms this suggestion, while the amount is too small to be observable in UV-Raman spectroscopy, which is comparably surface sensitive. Therefore, it is likely that a MoO_x concentration in the range of monolayers is present, but the exact amount is hard to quantify. To address that point we added a sentence to the revised manuscript on page 8.

REVIEWERS' COMMENTS:

Reviewer #1 (Remarks to the Author):

I have now read the rebuttal letter by the authors (and related answers to my initial comments) as well as the revised article and based on this material I am recommending the work for publication.

Reviewer #2 (Remarks to the Author):

The authors have adequately responded to my concerns and revised the manuscript and SI accordingly. I recommend publication of the manuscript in its present form.

Reviewer #3 (Remarks to the Author):

I am happy to see that the authors carefully considered all my comments and discussed their reasoning in the reply letter. Therefore, I recommend that the paper is published as it is.